# Dinosaurs: Comparative Cytogenomics of Their Reptile Cousins and Avian Descendants

**DOI:** 10.3390/ani13010106

**Published:** 2022-12-27

**Authors:** Darren K. Griffin, Denis M. Larkin, Rebecca E. O’Connor, Michael N. Romanov

**Affiliations:** 1School of Biosciences, University of Kent, Canterbury CT2 7NJ, UK; 2Department of Comparative Biomedical Sciences, Royal Veterinary College, University of London, London NW1 0TU, UK

**Keywords:** dinosaurs, birds, reptiles, chromosome, karyotype, cytogenomics, comparative genomics, genome evolution

## Abstract

**Simple Summary:**

Dinosaurs have been in scientific and popular culture since early fossil discoveries, but increased interest, particularly in their genomes, is expanding. Birds are reptiles, specifically theropod dinosaurs, meaning that if we compare the genomes of related reptile relations, we can get an idea of what the extinct dinosaur genomes looked like. In all animals/plants/fungi, we think of genome organization in terms of chromosomes. Genes sit on chromosomes and each cell of each individual of each species has its own unique organization. Every gene is in exactly the same spot on each chromosome, organized like continents and islands, with the genes as the cities/towns/villages. All reptiles apart from crocodilians have both big and small chromosomes in their genomes but birds particularly so, like the Philippines or Polynesia. Birds have ~80 chromosomes (far more than most organisms) and this is very consistent in most species. Recent studies suggest that this pattern was probably established ~255 million years ago as it is also mostly present in some turtles. In other words, most dinosaurs probably had chromosomes (genome organization) like chickens or emus. In this paper, we present ideas of how this may have contributed to dinosaurs being so diverse in appearance and function.

**Abstract:**

Reptiles known as dinosaurs pervade scientific and popular culture, while interest in their genomics has increased since the 1990s. Birds (part of the crown group Reptilia) are living theropod dinosaurs. Chromosome-level genome assemblies cannot be made from long-extinct biological material, but dinosaur genome organization can be inferred through comparative genomics of related extant species. Most reptiles apart from crocodilians have both macro- and microchromosomes; comparative genomics involving molecular cytogenetics and bioinformatics has established chromosomal relationships between many species. The capacity of dinosaurs to survive multiple extinction events is now well established, and birds now have more species in comparison with any other terrestrial vertebrate. This may be due, in part, to their karyotypic features, including a distinctive karyotype of around *n* = 40 (~10 macro and 30 microchromosomes). Similarity in genome organization in distantly related species suggests that the common avian ancestor had a similar karyotype to e.g., the chicken/emu/zebra finch. The close karyotypic similarity to the soft-shelled turtle (*n* = 33) suggests that this basic pattern was mostly established before the Testudine–Archosaur divergence, ~255 MYA. That is, dinosaurs most likely had similar karyotypes and their extensive phenotypic variation may have been mediated by increased random chromosome segregation and genetic recombination, which is inherently higher in karyotypes with more and smaller chromosomes.

## 1. Introduction

The question of the origin of reptiles, birds and their relationship to extinct dinosaurs has challenged many generations of biologists; it also continues to interest the lay public. In recent years, this interest has increased due to new paleontological findings and developments in the field of genomics (e.g., [1,2]). In light of recent paleontological findings, the hypothesis that dinosaurs were completely eradicated by the most recent mass extinction event [3,4] has been pervasive in the scientific literature, as well as fiction, film, television, popular culture and the media. However, this scientific dogma has undergone a fundamental revision in recent times; dinosaurs are now thought to be reptile survivors of the most recent extinction event through their evolution into modern birds (e.g., [1,2]). In other words, birds are both reptiles and dinosaurs.

A karyotype represents a map of the genome of interest and every genome sequence assembly would benefit from an accurate cytogenomic map [5]. However, while we can do this directly in extant species by sampling live material, the chromosomal composition of extinct dinosaurs can only be derived by inference. This conclusion can be reached by examining the whole genome chromosome-level assemblies (CLA) of extant species [5]. With information about several species’ CLAs at our disposal, comparative genomics is much more practical in silico [6]. While the auxiliary method of cross-species fluorescence in situ hybridization (zoo-FISH) can uncover further chromosome rearrangements that are difficult to detect using conventional karyotyping (e.g., [7,8,9,10]), comparative genomics enables us to outline the genome structure of less well-studied species (e.g., [8,11,12]) and reveal the chromosome rearrangements that led to each species’ distinct karyotype (e.g., [10]) using a reference species as a benchmark, e.g., chickens. The relevance and genomic correlates of such chromosome constituents as evolutionary breakpoint regions (EBRs) and homologous synteny blocks (HSBs) that are features of chromosome evolution [6], as well as the mechanisms behind chromosomal breakage and fusion, can all be addressed with the use of CLAs. Given the prevalence of genomics in modern scientific enquiry, cytogenetics (or, more precisely, cytogenomics) is not only a descriptive discipline but also offers a conceptual framework for the organization of any genome. It also provides an original framework for delineating genome–phenome relationships.

Aligning genome assemblies with the respective sets of chromosomes for the majority of species is a challenging task requiring different technologies. The difficulty in developing a genomic roadmap in birds is that the small microchromosomes belie the accurate identification with even contemporary methodologies. Herewith, cytogenomics specialists can examine CLAs in a wide range of bird species, offering new information about the genome structure of the extinct dinosaurs, the ancestors of modern birds and phylogenetic “cousins” of other extant reptiles.

## 2. Reptilia: Their Phylogeny and Karyotypes

The crown group Reptilia [13] incorporates extinct and existing clades of reptiles, dinosaurs and birds (Figure 1). In particular, it encompasses the diapsid reptiles including the Lepidosauria (tuatara, lizards and snakes) and the Archosauria (extinct dinosaurs, pterosaurs, crocodilians and birds); the latter having originated ~250 million years ago (MYA) [14]. The divergence of synapsids (mammals and their extinct ancestors) in one branch, and anapsids (turtles) and diapsids (other reptiles and birds) in the other, occurred about 310–350 MYA. Evolutionarily, birds represent a monophyletic group of homoeothermic reptiles and are believed to have arisen from theropod dinosaurs about 150 MYA (e.g., [14,15,16]). *Archaeopteryx* discovered from the late Jurassic (~150 MYA) is recognized as one of the earliest birds. Fossils of most orders of modern birds appear in the early part of the Cenozoic era (65–0 MYA). According to mitochondrial DNA comparisons with extant reptiles, birds are most closely linked to crocodilians, and the divergence between the two lineages is thought to has happened between 210 and 250 MYA (reviewed in [17]). The order Testudines (turtles, tortoises and terrapins) were separated from the Lepidosauria and the Archosauria in the traditional phylogeny because they were thought to be the only survivors of a presumed early anapsid reptile group. The results from molecular phylogeny data estimated from the nucleotide sequences of complete mitochondrial genomes and nuclear genes suggest that turtles should be grouped within the Archelosauria along with crocodilians and birds, while squamates (scaled reptiles including snakes and lizards) are classified into a different clade of Lepidosauria (e.g., [17,18,19,20,21]; Figure 1).

Similar to birds, two main chromosomal components of the karyotypes of snakes, turtles, lizards and tuatara (but not crocodilians) are the macro- and microchromosomes. Snakes have a limited spectrum of karyotypic variation. The diploid number 2*n* = 36, including 8 pairs of macro- and 10 pairs of microchromosomes, is the most prevalent karyotype in snakes (reviewed in [17]). Lizards also have a low karyotypic variation, mostly with 32–44 chromosomes (e.g., [26,27]) and the extremes being 16 [28] and 62 chromosomes [29]. The diploid number in the lizard *Anolis monticola* is 48 chromosomes, including 24 macro- and 24 microchromosomes. The fission of chromosomes has been demonstrated in conjunction with lower diploid numbers [30]. The karyotype of the indigenous New Zealand lizard genus *Sphenodon* (tuatara) has not changed for at least one million years. It has 36 chromosomes, with 14 pairs of macrochromosomes and 4 pairs of microchromosomes. The similarity of the karyotypes of *Sphenodon* and most Testudines (turtles) points to an ancestral karyotype with a complement of 14 pairs of macrochromosomes and varying numbers of microchromosome pairs [31]. The chromosome number of most crocodilians has long been known [32]; the American alligator (*Alligator mississippiensis*) karyotype (2*n*) consists of 32 macrochromosomes, but notably no microchromosomes (in contrast to other reptiles including birds, e.g., [33]). This peculiar feature, unique among reptiles, suggests a derived karyotype arising as a result of wholesale microchromosomal fusion, probably of single origin (given the small number of monophyletic species in which it is observed). Why crocodilians underwent this change and other reptiles did not is unclear.

Using cDNA clones of functional reptile genes and zoo-FISH, Matsuda et al. [17] created comparative cytogenetic maps of the Japanese four-striped rat snake (*Elaphe quadrivirgata*) and the Chinese soft-shelled turtle (*Pelodiscus sinensis*). The six biggest chromosomes were found to be near-identical between the chicken and turtle, indicating that chromosome homology was well conserved between the two species. However, compared to the turtle, the snake’s homology to the chicken chromosomes is lower. The chicken Z chromosome shares conserved synteny with the turtle 6q and the snake 2p chromosomes. These findings imply that conserved sequence blocks have survived during the evolution of Testudines and Archosauria in the genomes of turtles and birds. The lineage of snakes has a karyotype with a number of large-sized macrochromosomes and fewer microchromosomes due to a greater frequency of interchromosomal rearrangements that happened between the macrochromosomes and also between macro- and microchromosomes [17]. The suggested that the molecular phylogenetic links between the three genera are supported by the higher conserved synteny in the comparison between the chicken and turtle than in the comparison between the chicken and snake [18,20].

In the 2000s, bacterial artificial chromosome (BAC) libraries became available for the genomes of five reptilian species, American alligator (*Alligator mississippiensis*), garter snake (*Thamnophis sirtalis*), tuatara (*Sphenodon punctatus*), painted turtle (*Chrysemys picta*) and gila monster (*Heloderma suspectum*), which represent all five major lineages of extant reptiles [33,34]. The green anole lizard (*Anolis carolinensis*) was the first reptilian target species for which the genome sequence and CLA were produced [35], with the painted turtle [36], American alligator [37], garter snake [38] and a variety of other reptile species having followed. These advances, along with the progress in avian genomics, make it possible to study the evolutionary relationships and genome history of higher vertebrates (reptiles, birds and mammals) in a broader context [39]. Comparative mapping of birds and reptiles sheds additional light on the amniotes’ evolutionary history [17].

## 3. Defining Dinosaurs

According to *Britannica* [40], dinosaurs are described as “*Triceratops*, contemporary birds, their most recent common ancestor and all of their descendants.” However, for biologists, it could be simpler to picture dinosaurs as reptiles with hind limbs held erect beneath the trunk, similar to how mammals’ hind limbs are held. This sets dinosaurs apart from the majority of other reptiles, including lizards and crocodilians, whose legs are often placed to the side. The related evolutionary clades of dinosaurs, birds and reptiles within the crown group Reptilia [13] are shown in Figure 1. Dinosaurs can therefore be straightforwardly discerned from other animals if its easily identifiable sidelong sister branch of pterosaurs is taken out. With this in mind, dinosaurs are survivors of many extinction events including the most recent Cretaceous–Paleogene (K–Pg) [4]. Data combined from molecular cytogenetics and bioinformatics help demonstrate that their adaptability and capacity to survive extinction events may be due, at least in part, to their karyotypic features.

## 4. Dinosaurian Forefathers and Avian Heirs

The amniote lineage divided into the reptile/bird lineage (diapsids) and the synapsids, which eventually evolved into mammals (and others), ~325 MYA. Over 17,500 diapsid species exist on the planet, the majority of which are birds (~11,000 species). Turtles (Testudines) diverged first (~255 MYA), followed by crocodilians (~ 252 MYA), pterosaurs (~245 MYA) and then true dinosaurs (including birds) ~240 MYA [41,42]. All of these organisms, including dinosaurs and birds, share a common ancestor (Figure 1) that lived 275 MYA. Dinosaur species remained few in number for the following 30 million years, but during the Jurassic period, their numbers, geographic range and body sizes all increased [43]. The subsequent 135 million years of dinosaur evolution were remarkable because they were the dominant vertebrates on Earth and manifested an extraordinary diversity of species [1]. Amazingly, the dinosaurs survived the catastrophic extinction events of the Carnian–Norian and end-Triassic eras (228 and 201 MYA, respectively). There are currently more than 1000 known species of fossil, with around 30 new species (excluding birds) added each year [44].

Usually, the wide diversity and species abundance of dinosaurs is attributed to the extinction of competing species, which allowed the dinosaurs to prosper. However, it has also been suggested that these remarkable levels of abundance and diversity were a result of dinosaur-specific genetic adaptations, which let them outlive other species in hostile habitats. Examples include unusual bone development rates and highly adapted respiration systems [45], such as unidirectional respiration [46]. Avian species may have evolved successfully due to these types of adaptations; evidence for this may be found in the organization and structure of their genomes.

Multiple bird genome sequencing projects have corrected the important dates of avian diversification, thanks to a revised avian phylogeny based on genome assemblies [47,48]. When the Neognathae (Galloanserae/Neoaves) and the Palaeognathae (Ratites/Tinamous) split apart, this was the time of the first bird evolutionary divergence occurring around 100 MYA. The second divergence occurred when the Galloanserae (Galliformes and Anseriformes) and the Neoaves split 80 MYA, with the divergence of the Galliformes (landfowl, such as chicken, turkey, quail and pheasant) and the Anseriformes (waterfowl, i.e., geese, ducks and swans) occurring around 66 MYA. A further significant split of the Neoaves into the Columbea (including pigeons) and the Passerea (including songbirds) was earlier in evolutionary time (67–69 MYA). Around the time of these two major divergences and after the K–Pg mass extinction event [3,4], a total of 36 neoavian lineages evolved due to diversification in a very brief evolutionary period of 10–15 million years, as shown by Jarvis et al. [48] and Prum et al. [49]. Thus, comparative studies using genomics have revised our understanding of the evolution of dinosaurs, providing fascinating insights into the diversification and the evolution of phenotype [47,48], and prompting further research of the dinosaur karyotype.

## 5. Characterizing a Hypothetical Dinosaur Genome Organization

With no intact DNA available from dinosaur fossils, researchers can infer information about extinct dinosaur karyotypes by studying enough avian and reptile CLAs. Romanov et al. [50] were able to determine the most likely ancestral karyotype of all birds by aligning (near) chromosome-level assemblies from six extant birds and an outgroup of the *Anolis* lizard. This research strategy revealed that the common avian ancestor had a karyotype comparable to that of a chicken or ratite bird [1,50], being a bipedal, terrestrial, tiny Jurassic dinosaur with some flight capacity [1,51]. The next step was to retrace the most likely sequence of rearrangement occurrences that resulted in the avian species’ characteristic karyotypes (e.g., [10]). The zebra finch (*Taeniopygia guttata*) and budgerigar (*Melopsittacus undulatus*) were likely subject to the most intra- and interchromosomal changes, while the reconstructed ancestral genome makeup was actually closest to the common chicken karyotype among the birds explored [1,50]. Damas et al. [52] used the method DESCHRAMBLER on fragmented genome assemblies to rebuild the ancestral avian karyotype. A thorough examination of the structure of primitive avian chromosomes was conducted around 14 significant nodes in the evolution of birds. These findings elucidated the varying rates of rearrangement that took place throughout bird evolution. Additionally, it enabled the identification of patterns in the distribution of EBRs along the micro- and macrochromosomes.

A similar method was used by O’Connor et al. [53] to reproduce the diapsids’ most likely ancestral karyotype. A universally hybridizing BAC FISH probe set was created for this purpose [10], which was capable of directly hybridizing across species that diverged hundreds of millions of years ago [54]. The BAC probes used in zoo-FISH investigations produced distinctive signals on the chromosomes of anole lizard (*Anolis carolinensis*) and further on those of the red-eared slider (*Trachemys scripta*) and spiny soft-shelled turtle (*Apalone spinifera*). Based on these zoo-FISH examinations, the chromosome rearrangement events might then be anchored from the viewpoint of an ancestral archelosaur (bird–turtle). The chromosomal modifications from the diapsid ancestor through the archelosaur ancestor [55] and the theropod lineage, and to birds, including chickens, were thus recreated by merging molecular cytogenetics with bioinformatics data [1].

In addition to detecting macro- and microchromosomal homologues, the hybridization of BACs to *Trachemys scripta* (2*n* = 50) and *Anolis carolinensis* (2*n* = 36) metaphases also revealed the ancestral diapsid karyotype (275 MYA) with 2*n* = 36–46 and with the ratio of macro- to microchromosomes being approximately 1:1 [1,35,56]. The majority of the key characteristics linked to a typical bird karyotype were already set in the archelosaur progenitor 255 MYA [1,57], which experienced rapid transformation in the preceding 20 million years. We know this because the majority of the *Apalone spinifera* (2*n* = 66) and chicken (i.e., ancestral avian) chromosomes (numbered 1-28 + Z) are perfectly syntenic [1]. Studies using chicken chromosome painting on the chromosomes of the painted turtle (*Chrysemys picta*) [58], red-eared slider (*Trachemys scripta*; both 2*n* = 50) [9] and Chinese soft-shelled turtle (*Pelodiscus sinensis*; 2*n* = 66) [17] further support the hypothesis that macrochromosomes of birds and turtles are syntenic. Given this information, the only parsimonious explanation is that birds and *Pelodiscus sinensis* share a common ancestor in terms of their karyotypic structure, as the number of independent convergent events to achieve the same pattern would be statistically extremely unlikely.

To achieve the common avian karyotype pattern from this (~255 MYA) common archelosaur ancestor, to that present in the majority of the main groups of birds, including the Ratites, Galliformes, Anseriformes, Columbea, Passeriformes and others, only about seven fissions would be required. At the rate of chromosomal change occurring at the time, a complete bird-like karyotype would have most likely formed prior to the emergence of the earliest dinosaurs and pterosaurs ~240 MYA. That is, if the same fission rate that had been present for the preceding 20 million years was maintained for another 15 million years, the early dinosaurs probably had bird-like karyotypes [1,59].

The data available therefore strongly imply that not only in most birds, but also with a high degree of certainty, in many, if not most, extinct dinosaurs, the avian chromosomal pattern was maintained mostly unchanged [60]. Figure 2 illustrates this.

## 6. Further Insights into Karyotype Evolution

It had already been suggested that the genome of avian ancestors dating back to more than 80 MYA already had microchromosomes [61,62]. O’Connor et al. [53] asserted that this karyotype organization existed far earlier. They also disputed the idea that the fragmented genome organization (i.e., a karyotype with 2*n* ≈ 80 chromosomes) accompanied the genome size decrease in birds that has occasionally been linked to the evolution of flight. In other words, a certain correlation was previously thought to exist between genomes with fewer chromosomes (and no microchromosomes) and greater genome sizes (2.5–3 Gb), e.g., in mammals and crocodilians [37,63]. However, O’Connor et al. [62] hypothesized that the bird-like karyotype evolved first, followed by a decrease in genome size and then by the evolution of flight.

In theory, there are two potential reasons why a near-identical karyotype pattern has persisted for ~255 million years: either there is minimal opportunity for change, or the arrangement has been so successful in driving evolution that there is no need to alter. For the former, interchromosomal rearrangement, which is frequently observed in mammals but almost never observed in avian species, is facilitated by repeated elements. This implies that the lack of recombination hotspots [64,65], repeat structures [66,67,68] or endogenous retroviruses [50,69,70] in the avian karyotypes limit the options for interchromosomal rearrangement. Additionally, purifying selection acting on some of the smallest microchromosomes was demonstrated by Damas et al. [52]. However, a karyotype with little variation over 255 million years also suggests that it is an evolutionary success. The significant degree of phenotypic variation that we observe in dinosaurs (including birds) may be caused by the high rates of chromosome recombination and large number of chromosomes, particularly microchromosomes, [53]. That is, this variation is likely mediated by random chromosome segregation and increased genetic recombination. Although the presence of numerous chromosomes is by no means the sole means via which variation can be created, it may help to explain the apparent paradox of the dinosaurs’ enormous phenotypic diversity but low karyotypic diversity. Phenotypic variation is the driving force behind evolution. O’Connor et al. [53] acknowledged the possibility/likelihood that some dinosaurs underwent a significant amount of interchromosomal alteration. Modern instances include multiple fusions in parrots [7,71], falcons [53,72] and many fissions in kingfishers [73]; see Figure 2. It may never be known which particular extinct dinosaur groups accomplished this, if any.

## 7. What Else Can Be Learned from Cytogenomics?

The main mechanism for chromosomal change in the evolution of the dinosaur genome was likely chromosome inversion with few or no interchromosomal rearrangements. Contiguous ancestral regions (CARs), which are most likely to reflect the chromosomes of the diapsid ancestor, were described by Griffin et al. [1] using the ancestral genome reconstruction program Multiple Genome Rearrangement and Analysis (MGRA) [74]. Although this number was likely underestimated, 49 inversions along the route from the diapsid progenitor to the present chicken were found [1]. Even in chickens, the rate of intrachromosomal alteration may have accelerated in modern times [50]. However, several bird clades, especially the songbirds, the group with the greatest number of species, showed an even larger degree of rearrangement [47,70,75]. The possibility that periods of rapid speciation may have also coincided with higher rates of chromosomal inversion in other dinosaur groups seems plausible [50,62,75].

Around 400 HSBs flanked by EBRs that define the evolution of the dinosaur genome were discovered by O’Connor et al. [62]. The EBRs frequently exist in gene-dense regions with genes involved in lineage-specific biology, transposable elements and other repetitive sequences, according to prior genomic studies in other species (mainly mammals) [76,77,78,79,80]. In contrast, HSBs have a greater number of regulatory and developmental genes [67,77]. Chromosome breaks that damage important genes or do not offer a selective benefit are more likely to be repaired in populations even if found in regions prone to breakage, such as open chromatin regions or recombination hotspots [70].

Using gene ontology (GO) methods, substantial enrichments in the HSB regions were determined with respect to the genes responsible for the development of sensory organs, amino acid transmembrane transport and signaling, plus synapse/neurotransmitter transport, nucleoside metabolism, cell morphogenesis and cytoskeleton [62]. The dinosaur findings reported by O’Connor et al. [62] corroborate the concept that HSBs are enriched for GO terms associated with evolutionary constant phenotypic traits [79]. One such feature is the *Hox* code and its relevance to facilitating both the species diversity and evolutionary success of tetrapods [81]

On the other hand, EBRs are frequently suggested as active spots in genome evolution [82]. In avian EBRs, we observe enrichment for GO terms pertaining to certain adaptation traits, such as forebrain development in budgerigar EBRs (relevant to vocal learning) [70]. There are other significant enrichments of EBRs with genes and single GO terms related to chromatin modification, chromosomal architecture and proteasome/signalosome structure [62]. Chromosomal rearrangements arise, mechanistically, as a result of genetic recombination, DNA repair and/or replication mechanisms occurring following double-strand breaks or replication fork breakage/stalling. Non-Allelic Homologous Recombination (NAHR), Non-Homologous End-Joining (NHEJ), Fork Stalling and Template Switching (FoSTeS) and Microhomology-Mediated Break-Induced Replication (MMBIR) are all mechanisms implicated in this process. The importance of high-resolution analysis in determining the DNA sequence around EBRs has been highlighted [83] as a means of unravelling which specific mechanisms are involved. Transposable elements are also a source of diverse cis-regulatory sequences, constituting a large part of eukaryotic genomes [84]. As our understanding of the biological influence of genomic transposable elements increases, their relevance in chromosomal change, especially in birds is becoming apparent. Flighted birds appear to have smaller genomes yet more transposable elements than their flightless counterparts and all display genomic instability in the genomic regions that are enriched for these sequences [85]. Integrating cytogenomics, single-molecule technologies and genome assemblies in which the repetitive elements have been properly defined is therefore essential to understand how and why the genomes of birds (and their forebears, the extinct dinosaurs) have evolved.

Two recent studies [86,87] introduced a new concept in the context of genomic rearrangements affecting regulation in constantly evolving systems—that of topologically associating domains (TADs). The TADs are conserved inter-species; they buffer evolutionary rearrangements and conserve long-range interactions. Surprisingly, they nonetheless often span EBRs in close proximity to genes with species-specific expression (e.g., in immunological cells). They thus generate novel enhancer-promoter interactions exclusive to the species of interest. In other words, the TAD boundaries are disrupted by EBRs and enable sequence-conserved enhancer elements (from various locations in the genome) inter-species to create unique regulatory modules [86]. All animal genomes are thought to be sequestered into TADs, and they also insulate gene promoters from enhancers. Evolutionary chromosome rearrangements disrupt TAD structure and thereby generate novel regulatory interactions between promoters and enhancers that were historically physically separated. In turn, this could lead to new genomic expression patterns. These could cause deleterious phenotypes but could, nonetheless, create patterns and phenotypes that are evolutionarily advantageous. The EBRs therefore may influence TAD structure in the context of the evolution of gene regulation and of phenotypes of the various different species that arise [87]. No doubt attention will turn to TADs in the study of reptilian chromosome evolution in the coming years.

## 8. Conclusions

The finding that the avian-like karyotype probably predates the appearance of dinosaurs adds to the paleontological research showing that feathers and pneumatized skeletons initially appeared in more ancient dinosaur or archosaurian forebears [58,82]. For 200 million years, dinosaurs dominated the animal kingdom, with substantial radiations following two great extinction events. Plasticity of the dinosaur clade (including modern birds) in terms of remarkable variation and number of species [88] is noticeable in spite of the near eradication after the K–Pg extinction event [4].

In comparison with other established methodologies, the cytogenomic examination of the possible dinosaur karyotype shines new light on genome evolution, with insights regarding phenotype and an alternate avenue of inquiry [89]. In this regard, this is much more than just a curation effort or a conjectural one. The recent studies outlined and discussed here have shown a peculiar paradox of the dinosaur genome structure that is quite possibly the cause of such phenotypic evolutionary change yet being strikingly karyotypically unchanged in the course of evolution.

## Figures and Tables

**Figure 1 animals-13-00106-f001:**
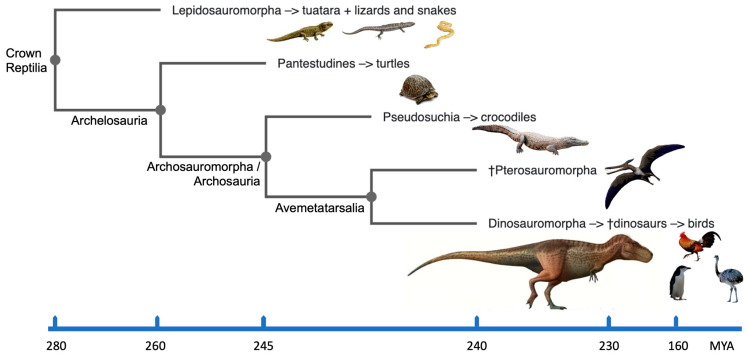
A simplified cladogram of the crown group Reptilia for major evolutionary groups including dinosaurs, birds and reptiles, based on [22,23,24] and plotted using the Phylo.io webtool [25]. † extinct groups. Time scale is not linear.

**Figure 2 animals-13-00106-f002:**
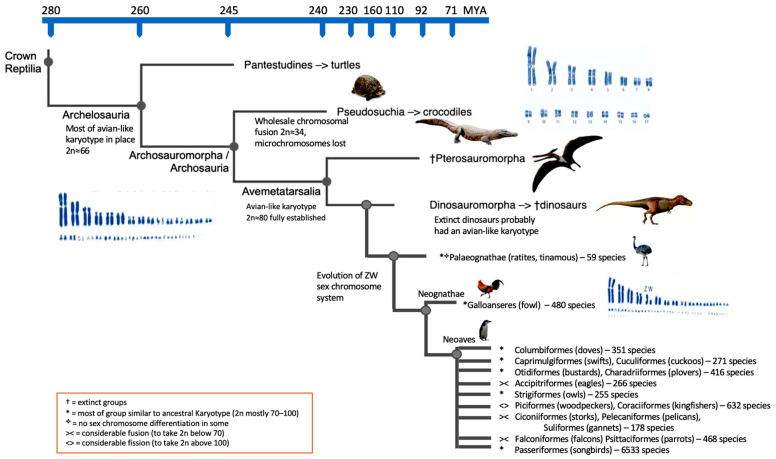
Cladogram of the major evolutionary reptilian groups including dinosaurs and several groups of birds. Likely karyotypic changes given, time scale is not linear.

## Data Availability

Not applicable.

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
