# Peer review of "Dinosaurs: Comparative Cytogenomics of Their Reptile Cousins and Avian Descendants"

_animals, 2022, doi:10.3390/ani13010106_

Round 1
Reviewer 1 Report
The manuscript submitted by Griffin et al. attempts to review the state of the art on the possible karyotype of dinosaurs (including birds) and the importance of its structure in mediating rapid evolution and adaptive radiation. While the topic is of great interest for both the study of Evolution and Developmental Biology (jointly Evo-Devo), this review has important shortcomings and flaws, which are summarized as follows:
- The topic should be further discussed, including important aspects such as the evolutionary conservation of genes involved in animal morphology [1] and repeated elements, the latter being involved in chromosomal rearrangements [2] and in the evolution of cis-regulatory elements of gene expression [3]. In fact, the involvement of transposons (a type of repeat element) in avian genome evolution is already known, which is not mentioned in the review by Griffin et al [4].
- A more formal language, appropriate for a scientific document, should be used. For example, it is inappropriate to talk about the personal preferences of the director of Jurassic Park, Steven Spielberg. Colloquial expressions should also be avoided. Please also beware of typos (e.g., in the sentence on lines 35-36 the verb is missing).
- Some sections of the review are not very relevant and repeat the same statements as in other sections. These repetitions artificially and unnecessarily extend the document. Please consider the readers: a short and straightforward, but complete review is better.
- The original sources, not reviews containing them, should be cited whenever allusion is made to specific data from these studies. There are also numerous citations to a previous review, also written by Griffin et al. (see the first reference in the manuscript). I consider this to be an unjustified self-citation.
- There is a lack of figures that show similarities in the karyotypic structure of different groups of reptiles (including birds) and give a graphical explanation of likely karyotypic changes throughout the evolution of birds and maybe other dinosaurs.
- Figure 1 seems to be of low relevance. What useful information does it provide? Not even a scale on the y-axis is indicated, nor is the content of each panel adequately explained. On the other hand, the cladogram in Figure 2 should include an estimate of how long ago each of these reptile groups diverged.
- Why do crocodiles, despite being the most closely related group of extant reptiles to birds, not have the same karyotypic structure? It is assumed that this structure was already present in the common ancestor with the soft-shelled turtles. Moreover, why, if soft-shelled turtles have the same karyotypic structure, do they not have as much phenotypic variability as dinosaurs?
References:
[1] Böhmer, C., Rauhut, O. W., & Wörheide, G. (2015). Correlation between Hox code and vertebral morphology in archosaurs. Proceedings of the Royal Society B: Biological Sciences, 282(1810), 20150077.
[2] Burssed, B., Zamariolli, M., Bellucco, F. T., & Melaragno, M. I. (2022). Mechanisms of structural chromosomal rearrangement formation. Molecular Cytogenetics, 15(1), 1-15.
[3] Sundaram, V., & Wysocka, J. (2020). Transposable elements as a potent source of diverse cis-regulatory sequences in mammalian genomes. Philosophical Transactions of the Royal Society B, 375(1795), 20190347.
[4] Kapusta, A., & Suh, A. (2017). Evolution of bird genomes—a transposon's‐eye view. Annals of the New York Academy of Sciences, 1389(1), 164-185.
Author Response
Dear Editors – please see responses to reviewer comment below:
Reviewer 1
Comments and Suggestions for Authors
The manuscript submitted by Griffin et al. attempts to review the state of the art on the possible karyotype of dinosaurs (including birds) and the importance of its structure in mediating rapid evolution and adaptive radiation. While the topic is of great interest for both the study of Evolution and Developmental Biology (jointly Evo-Devo), this review has important shortcomings and flaws, which are summarized as follows:
REPLY: We are sorry that the reviewer thinks that there are shortcomings and flaws. They are easily addressed (below). All corrections appear in red in the new manuscript
- The topic should be further discussed, including important aspects such as the evolutionary conservation of genes involved in animal morphology [1] and repeated elements, the latter being involved in chromosomal rearrangements [2] and in the evolution of cis-regulatory elements of gene expression [3]. In fact, the involvement of transposons (a type of repeat element) in avian genome evolution is already known, which is not mentioned in the review by Griffin et al [4].
REPLY: We have added a section discussing this based on the references below. Lines 323-325 and 331-347
- A more formal language, appropriate for a scientific document, should be used. For example, it is inappropriate to talk about the personal preferences of the director of Jurassic Park, Steven Spielberg. Colloquial expressions should also be avoided. Please also beware of typos (e.g., in the sentence on lines 35-36 the verb is missing).
REPLY: Verb added on line 35. Text made more formalized as requested (red text throughout)
- Some sections of the review are not very relevant and repeat the same statements as in other sections. These repetitions artificially and unnecessarily extend the document. Please consider the readers: a short and straightforward, but complete review is better.
REPLY: We have shortened, where we feel there is repetition and perceived irrelevance. Throughout manuscript
- The original sources, not reviews containing them, should be cited whenever allusion is made to specific data from these studies. There are also numerous citations to a previous review, also written by Griffin et al. (see the first reference in the manuscript). I consider this to be an unjustified self-citation.
REPLY: Self-citation removed, and replaced as requested (reference 1). All other references contain predominantly original data as requested. When reviews are included, it is where they contain broader statements relevant to the narrative of the review.
- There is a lack of figures that show similarities in the karyotypic structure of different groups of reptiles (including birds) and give a graphical explanation of likely karyotypic changes throughout the evolution of birds and maybe other dinosaurs.
REPLY: See the new figure 2.
- Figure 1 seems to be of low relevance. What useful information does it provide? Not even a scale on the y-axis is indicated, nor is the content of each panel adequately explained. On the other hand, the cladogram in Figure 2 should include an estimate of how long ago each of these reptile groups diverged.
REPLY: Old figure 1 removed. Old figure 2 (now figure 1) - timeline added.
- Why do crocodiles, despite being the most closely related group of extant reptiles to birds, not have the same karyotypic structure? It is assumed that this structure was already present in the common ancestor with the soft-shelled turtles. Moreover, why, if soft-shelled turtles have the same karyotypic structure, do they not have as much phenotypic variability as dinosaurs?
REPLY: It is not known why crocodiles have the same karyotypic structure – this is now mentioned on lines 122-125. The two soft shelled turtles with similar karyotypes to birds (Chinese and spiny) represent only two closely related species among many. Turtles, as a group, do not have avian like karyotypes. A single species (or two closely related ones) would not be expected (by definition) to have inter-specific species diversity. The issue of the evolution of the testudine-bird common ancestral karyotype
References:
[1] Böhmer, C., Rauhut, O. W., & Wörheide, G. (2015). Correlation between Hox code and vertebral morphology in archosaurs. Proceedings of the Royal Society B: Biological Sciences, 282(1810), 20150077.
[2] Burssed, B., Zamariolli, M., Bellucco, F. T., & Melaragno, M. I. (2022). Mechanisms of structural chromosomal rearrangement formation. Molecular Cytogenetics, 15(1), 1-15.
[3] Sundaram, V., & Wysocka, J. (2020). Transposable elements as a potent source of diverse cis-regulatory sequences in mammalian genomes. Philosophical Transactions of the Royal Society B, 375(1795), 20190347.
[4] Kapusta, A., & Suh, A. (2017). Evolution of bird genomes—a transposon's‐eye view. Annals of the New York Academy of Sciences, 1389(1), 164-185.
Reviewer 2 Report
It is a really interesting and complete review of the genetic relationship between dinosaurs and existing reptiles/birds. The manuscript is full of scientific information but easy-to-read and understand at the same time.
I have just a few little comments:
- L. 64: "this idea has undergone a drastic revision is recent times". I think "is" should be "in".
- I did not find the meaning of EBRs or HSBs in the text, please include them.
Author Response
Reviewer 2
Comments and Suggestions for Authors
It is a really interesting and complete review of the genetic relationship between dinosaurs and existing reptiles/birds. The manuscript is full of scientific information but easy-to-read and understand at the same time.
I have just a few little comments:
- L. 64: "this idea has undergone a drastic revision is recent times". I think "is" should be "in".
REPLY: Done
- I did not find the meaning of EBRs or HSBs in the text, please include them.
REPLY: They are defined on lines 67-68
Round 2
Reviewer 1 Report
Griffin et al. have mainly taken my corrections and recommendations into account in this revised version of the manuscript. Given that the main hypothesis underlying this review is that chromosomal rearrangements allowed for a great diversification of dinosaur morphologies and increased adaptability, I believe it is critical that the authors highlight possible alterations in topologically associated domains and promoter-enhancer interactions as one of the most plausible underlying mechanisms to explain this rapid evolution. The following references may be a good starting point to further explore this aspect:
Gilbertson, Sarah E., et al. "Topologically associating domains are disrupted by evolutionary genome rearrangements forming species-specific enhancer connections in mice and humans." Cell Reports 39.5 (2022): 110769.
Preger-Ben Noon, Ella, and Nicolás Frankel. "Can changes in 3D genome architecture create new regulatory landscapes that contribute to phenotypic evolution?." Essays in Biochemistry 66.6 (2022): 745-752.
I also found a typo (lines 30-31):
"The capacity of dinosaurs to survive multiple extinction events is now well established and birds now more species in comparison with any other terrestrial vertebrate." -> "The capacity of dinosaurs to survive multiple extinction events is now well established and birds now HAVE more species in comparison with any other terrestrial vertebrate."
Please review the manuscript carefully for possible additional typos.
Author Response
We have now added a section (see red text) on TADs